# Variance Estimation after Kernel Ridge Regression Imputation

**Hengfang Wang** [* 1]  **Jae-kwang Kim** [1]

## Abstract

Imputation is a popular technique for handling missing data. Variance estimation after imputation is an important practical problem in statistics. In this paper, we consider variance estimation of the imputed mean estimator under the kernel ridge regression imputation. We consider a linearization approach which employs the covariate balancing idea to estimate the inverse of propensity scores. The statistical guarantee of our proposed variance estimation is studied when a Sobolev space is utilized to do the imputation, where $\sqrt{n}$-consistency can be obtained. Synthetic data experiments are presented to confirm our theory.

## 1. Introduction

With the development of technology, massive data is more available and being analyzed by data analysts, which illuminates the way for big data era. Meanwhile, missing data issue has become more serious than before. For example, in web-based survey questionnaires, some people are reluctant to report their household incomes, while they are more willing to report other information such as education, gender, age group and so on. In practice, ignoring the cases with missing values can lead to misleading results (Kim & Shao, 2013; Little & Rubin, 2019).

Imputation is a popular technique for handling missing data. After imputation, the imputed dataset can then be served as a complete dataset with no missingness to be analyzed, which in turn makes results from different analysis methods consistent. However, treating imputed data as if observed and applying the standard estimation procedure may result in misleading inference, leading to underestimation of the variance of imputed point estimators. As a result, how to do statistical inference for the imputed estimator becomes an arising issue.

---

[*]Equal contribution [1]Department of Statistics, Iowa State University, Ames, IA 50011, USA. Correspondence to: Jae-kwang Kim <jkim@iastate.edu>.

*Presented at the first Workshop on the Art of Learning with Missing Values (Artemiss) hosted by the $37^{th}$ International Conference on Machine Learning (ICML).* Copyright 2020 by the author(s).

### 1.1. Related works

An overview of imputation method can be found in Haziza (2009). Multiple imputation, proposed by Rubin (2004), is a popular technique for imputation. However, the validity of its variance estimation needs to satisfy certain conditions (Fay, 1992; Wang & Robins, 1998; Kim et al., 2006; Yang & Kim, 2016). An alternative method is the fractional imputation, originally proposed by Kalton & Kish (1984). The main idea of fractional imputation is to generate multiple imputed values and the corresponding fractional weights. Hot deck imputation is a popular method of imputation where the imputed values are taken from the observed values. In this vein, Fay (1996); Kim & Fuller (2004); Fuller & Kim (2005); Durrant et al. (2005); Durrant & Skinner (2006) discussed fractional hot deck imputation. Further, Kim (2011) and Kim & Yang (2014) employed fully parametric approach to handling nonresponse items with fractional imputation. However, such parametric fractional imputation relies heavily on the parametric model assumptions. To mitigate the effects of parametric model assumption, empirical likelihood (Owen, 2001; Qin & Lawless, 1994) as a semi-parametric approach was considered. In particular, Wang & Chen (2009) employed the kernel smoothing approach to do empirical likelihood inference with missing values. Chen & Kim (2017) extended Müller et al. (2009)'s work to do fractional imputation with a regression model assumption. Cheng (1994) utilized the Kernel-based nonparametric regression approach to do the imputation and established the $\sqrt{n}$-consistency of the imputed estimator.

The kernel ridge regression (KRR) (Friedman et al., 2001; Shawe-Taylor et al., 2004) is a popular data-driven approach which can alleviate the effect of model assumption. With a regularized *M*-estimator in reproducing kernel Hilbert space (RKHS), KRR can capture the hidden model with complex RKHS while a regularized term makes the original infinite dimentional estimation problem viable (Wahba, 1990). van de Geer (2000), Mendelson (2002), Zhang (2005), Koltchinskii et al. (2006), and Steinwart et al. (2009) studied the error bounds for the estimates of RKHS-based method. Recently, Zhang et al. (2013) employed truncation analysis to estimate the error bound in a distributed fashion. Yang et al. (2017) considered randomized sketches for KRR and studied projection dimension which can preserve minimax optimal approximations for KRR.

## 1.2. Our Work

In this paper, we apply KRR as a nonparametric imputation method and develop a method to establish the consistent variance estimation for KRR imputation estimator under missing at random (MAR) assumption. To the best of our knowledge, this is the first paper which considers KRR technique and discusses its variance estimation in the imputation framework. Specifically, we first prove $\sqrt{n}$-consistency of the KRR imputation estimator and obtain influence function for linearization. After that, we employ a covariate balancing method (Wong & Chan, 2018) to get a valid estimate of the inverse of the propensity scores. The consistency of our variance estimator can thus be established. We additionally conducted a numerical experiment to confirm the validity of our proposed estimators.

## 2. Method

Consider the problem of estimating $\theta = \mathbb{E}(Y)$ from an independent and identically distributed (IID) sample of $(\boldsymbol{x}_i, y_i), i = 1, \cdots, n$. Instead of observing $y_i$, suppose that we observe $y_i$ only if $\delta_i = 1$, where $\delta_i$ is the response indicator function of unit $i$ taking values on $\{0, 1\}$. We assume that the response mechanism is MAR (Rubin, 1976) in the sense of

$$\delta \perp Y \mid \boldsymbol{x}.$$

Under MAR, we can develop a nonparametric estimator $\widehat{m}(\boldsymbol{x})$ of $m(\boldsymbol{x}) = \mathbb{E}(Y \mid \boldsymbol{x})$ and construct the following imputation estimator:

$$\widehat{\theta}_I = \frac{1}{n} \sum_{i=1}^{n} \left\{ \delta_i y_i + (1 - \delta_i)\widehat{m}(\boldsymbol{x}_i) \right\}. \tag{1}$$

If $\widehat{m}(\boldsymbol{x})$ is constructed by the Kernel-based nonparametric regression method, we can express

$$\widehat{m}(\boldsymbol{x}) = \frac{\sum_{i=1}^{n} \delta_i K_h(\boldsymbol{x}_i, \boldsymbol{x}) y_i}{\sum_{i=1}^{n} \delta_i K_h(\boldsymbol{x}_i, \boldsymbol{x})} \tag{2}$$

where $K_h(\cdot)$ is the Kernel function with bandwidth $h$. Under some suitable choice of the bandwidth $h$, Cheng (1994) first established the $\sqrt{n}$-consistency of the imputation estimator (1) with nonparametric function in (2).

In this paper, we extend the work of Cheng (1994) by considering a more general type of the nonparametric imputation, called Kernel ridge regression (KRR) imputation. The KKR technique can be understood using the reproducing Kernel Hilbert space (RKHS) theory (Aronszajn, 1950) and can be described as

$$\widehat{m} = \underset{m \in \mathcal{H}}{\arg\min} \left[ \sum_{i=1}^{n} \delta_i \left\{ y_i - m(\boldsymbol{x}_i) \right\}^2 + \lambda \|m\|_{\mathcal{H}}^2 \right], \tag{3}$$

where $\|m\|_{\mathcal{H}}^2$ is the norm of $m$ in the Hilbert space $\mathcal{H}$. Here, the inner product $\langle \cdot, \cdot \rangle_{\mathcal{H}}$ is induced by such kernel function, i.e.,

$$\langle f, K(\cdot, \boldsymbol{x}) \rangle_{\mathcal{H}} = f(\boldsymbol{x}), \forall \boldsymbol{x} \in \mathcal{X}, f \in \mathcal{H}, \tag{4}$$

namely, the reproducing property of $\mathcal{H}$. Naturally, such reproducing property implies the $\mathcal{H}$ norm of $f$: $\|f\|_{\mathcal{H}} = \langle f, f \rangle_{\mathcal{H}}^{1/2}$.

By the representer theorem for RKHS (Wahba, 1990), the estimate in (3) must lie in the linear span of $\{K(\cdot, \boldsymbol{x}_i), i = 1, \ldots, n\}$. Specifically, we have

$$\widehat{m}(\cdot) = \sum_{i=1}^{n} \widehat{\alpha}_{i,\lambda} K(\cdot, \boldsymbol{x}_i), \tag{5}$$

where

$$\widehat{\boldsymbol{\alpha}}_\lambda = (\boldsymbol{\Delta}_n \mathbf{K} + \lambda \mathbf{I}_n)^{-1} \boldsymbol{\Delta}_n \boldsymbol{y},$$

$\boldsymbol{\Delta}_n = \text{diag}(\delta_1, \ldots, \delta_n)$, $\mathbf{K} = (K(\boldsymbol{x}_i, \boldsymbol{x}_j))_{ij} \in \mathbb{R}^{n \times n}$, $\boldsymbol{y} = (y_1, \ldots, y_n)^{\text{T}}$ and $\mathbf{I}_n$ is the $n \times n$ identity matrix. Using the KRR imputation in (3), we aim to establish the following two goals:

1. Find the sufficient conditions for the $\sqrt{n}$-consistency of the imputation estimator $\widehat{\theta}_I$ using (5) and give a formal proof.

2. Find a linearization variance formula for the imputation estimator $\widehat{\theta}_I$ using the KRR imputation.

The first part is formally presented in Theorem 1 in Section 3. For the second part, we employ the covariate balancing idea of Wong & Chan (2018) to get a consistent estimator of $\omega(\boldsymbol{x}) = \{\pi(\boldsymbol{x})\}^{-1}$ in the linearized version of $\hat{\theta}_I$. By Theorem 1, we use the following estimator to estimate the variance of $\widehat{\theta}_I$ in (2):

$$\widehat{\mathbf{V}}(\widehat{\theta}_I) = \frac{1}{n(n-1)} \sum_{i=1}^{n} (\hat{\eta}_i - \bar{\eta})^2 \tag{6}$$

where

$$\hat{\eta}_i = \widehat{m}(\boldsymbol{x}_i) + \delta_i \widehat{\omega}_i \left\{ y_i - \widehat{m}(\boldsymbol{x}_i) \right\},$$

and $\widehat{\boldsymbol{\omega}} = (\widehat{\omega}_1, \ldots, \widehat{\omega}_n)^{\text{T}} \in \mathbb{R}^n$ is estimated by

$$\widehat{\boldsymbol{\omega}} = \underset{\boldsymbol{\omega} \geqslant 1}{\arg\min} \left[ \max_{u \in \widetilde{\mathcal{H}}_n} \left\{ S_n(\boldsymbol{\omega}, u) - \lambda \|u\|_{\mathcal{H}}^2 \right\} + \tau V_n(\boldsymbol{\omega}) \right]. \tag{7}$$

In (7), $S_n(\boldsymbol{\omega}, u) = \{n^{-1} \sum_{i=1}^{n} (\delta_i \omega_i - 1) u(\boldsymbol{x}_i)\}^2$ denotes the empirical validity measure for covariate-balancing between complete cases and all samples via function $u \in \widetilde{\mathcal{H}}_n = \{u \in \mathcal{H} : \|u\|_n = 1\}$, where $\|u\|_n^2 =$

**Algorithm 1** Variance Estimation for KRR Imputation

**Input:** observed a data set $\{(\delta_i, \boldsymbol{x}_i, \delta_i y_i)\}_{i=1}^n$, where vector length of $\boldsymbol{x}_i$ is $d$, $r = \sum_{i=1}^n \delta_i$ kernel function $K(\cdot, \cdot)$, regularization parameters $\lambda^{(t)}$, for $t = 1, \ldots, T$, regularization parameters $\tau^{(s)}$ for $s = 1, \ldots, S$, threshold parameter $\zeta$.

Select optimal $\widehat{\lambda}$ in (5) from $\{\lambda^{(t)}\}_{t=1}^T$ which minimizes the GCV criterion in (8).

**for** $s = 1$ **to** $S - 1$ **do**

    Optimize (7) with tuning parameters $\widehat{\lambda}$ and $\tau^{(s+1)}$ and get $\boldsymbol{\omega}^{(s+1)}$.

    **if** $B_n(\widehat{\boldsymbol{\omega}}^{(s+1)}) > \zeta$ **then**

        Set $\widehat{\tau} = \tau^{(s)}$.

        Stop the **for** loop.

    **end if**

**end for**

**Output**: variance estimator computed from (6) with $\widehat{\lambda}$ and $\widehat{\tau}$.

---

$n^{-1} \sum_{i=1}^n u(\boldsymbol{x}_i)^2$, and $V_n(\boldsymbol{\omega}) = n^{-1} \sum_{i=1}^n \delta_i \omega_i^2$ which serves as a penalizer to control the wiggliness of $\boldsymbol{\omega}$. The constraint $\boldsymbol{\omega} \geqslant 1$ indicates that $\omega_i \geqslant 1$ element-wisely. It is noted that we can set $\widehat{\omega}_i = 1$ where $\delta_i = 0$ and only optimize (7) on $\{\omega_i : \delta_i = 1\}$. Roughly speaking, the optimization problem (7) can be solved by limited-memory Broyden–Fletcher–Goldfarb–Shanno with bound constraints (L-BFGS-B) algorithm and computational details can be referred to Wong & Chan (2018). Unlike Wong and Chan (2018), we fixed the tuning parameter $\widehat{\lambda}$ selected via generalized cross-validation (GCV) in KRR in order to keep similar complexity of RKHS in both optimization problem and we only tune $\tau$, where the GCV criterion is

$$\text{GCV}(\lambda) = \frac{n^{-1} \|\{\boldsymbol{\Delta}_n - \mathbf{A}(\lambda)\} \boldsymbol{y}\|_2^2}{n^{-1} \text{Trace}(\boldsymbol{\Delta}_n - \mathbf{A}(\lambda))}, \quad (8)$$

and $\mathbf{A}(\lambda) = \boldsymbol{\Delta}_n \mathbf{K}(\boldsymbol{\Delta}_n \mathbf{K} + \lambda \mathbf{I}_n)^{-1} \boldsymbol{\Delta}_n$. For tuning parameter selection, define

$$B_n(\boldsymbol{\omega}) = \sup_{u \in \widetilde{\mathcal{H}}_n} \left\{ S_n(\boldsymbol{\omega}, u) - \lambda \|u\|_{\mathcal{H}}^2 \right\},$$

which measures the covariate balancing error. As $\tau$ increases, the corresponding $B_n(\widehat{\boldsymbol{\omega}})$ would increase. Take a sequence of $\tau$ as $0 < \tau^{(1)} < \ldots < \tau^{(S)}$, let $\widehat{\boldsymbol{\omega}}^{(s)} = \widehat{\boldsymbol{\omega}}(\tau^{(s)})$ be the minimizer of (7) given the tuning parameter $(\widehat{\lambda}, \tau^{(s)})$ for $s = 1, \ldots, S$. Our choice is $s^\star$ such that $s^\star$ is the largest in $\{1, \ldots, S\}$ such that $B_n(\widehat{\boldsymbol{\omega}}^{(s^\star)}) \leqslant \zeta$. We take $\zeta = 10^{-6}$ in our synthetic data analysis. The whole algorithm for imputation is summarized in Algorithm 1.

## 3. Theoretical Guarantee

We first make the following assumptions.

**Assumption 1.** *For some $k \geqslant 2$, there is a constant $\rho < \infty$ such that $\mathbb{E}[\phi_j(\mathbf{X})^{2k}] \leqslant \rho^{2k}$ for all $j \in \mathbb{N}$, where $\{\phi_j\}_{j=1}^\infty$ are orthonormal basis by expansion from Mercer's theorem.*

**Assumption 2.** *The function $m \in \mathcal{H}$, and for $\boldsymbol{x} \in \mathcal{X}$, we have $\mathbb{E}[\{Y - m(\boldsymbol{x})\}^2] \leqslant \sigma^2$, for some $\sigma^2 < \infty$.*

**Assumption 3.** *The propensity score $\pi(\cdot)$ is uniformly bounded away from zero. In particular, there exists a positive constant $c > 0$ such that $\pi(\boldsymbol{x}_i) \geqslant c$, for $i = 1, \ldots, n$.*

**Assumption 4.** *The ratio $d/\ell < 2$ for $d$-dimensional Sobolev space of order $\ell$, where $d$ is the dimension of covariate $\boldsymbol{x}$.*

The first assumption is a technical assumption which controls the tail behavior of $\{\phi_j\}_{j=1}^\infty$. Assumption 2 simply assumes that the noises have bounded variance. Assumption 1 and Assumption 2 together aim to control the error bound of the KRR estimate $\widehat{m}$. Further, Assumption 3 is used to make $\{\pi(\boldsymbol{x})\}^{-1}$ bounded above and it is a standard assumption in the missing data literature. Assumption 4 is a technical assumption for entropy analysis. Intuitively, when the dimension is large, the corresponding Sobolev space should be large enough to capture the true model.

**Theorem 1.** *Suppose Assumption $1 \sim 4$ hold for a Sobolev kernel of order $\ell$, $\lambda \asymp n^{1-\ell}$, we have*

$$\sqrt{n}(\widehat{\theta}_I - \widetilde{\theta}_I) = o_p(1), \quad (9)$$

*where*

$$\widetilde{\theta}_I = \frac{1}{n} \sum_{i=1}^n \left[ m(\boldsymbol{x}_i) + \delta_i \frac{1}{\pi(\boldsymbol{x}_i)} \{y_i - m(\boldsymbol{x}_i)\} \right] \quad (10)$$

*and*

$$\sqrt{n} \left( \widetilde{\theta}_I - \theta \right) \xrightarrow{\mathcal{L}} N(0, \sigma^2),$$

*with*

$$\sigma^2 = V\{E(Y \mid \boldsymbol{x})\} + E\{V(Y \mid \boldsymbol{x})/\pi(\boldsymbol{x})\}.$$

In Theorem 1, the notation $\asymp$ means that there exists constants $c_1 > c_2 > 0$, such that $c_2 n^{1-\ell} < \lambda < c_1 n^{1-\ell}$. Theorem 1 guarantees the asymptotic equivalence of $\widehat{\theta}_I$ and $\widetilde{\theta}_I$ in (10). Specifically, the reference distribution is a combination of outcome model and sampling mechanism. The variance of $\widetilde{\theta}_I$ achieves the semiparametric lower bound of Robins et al. (1994). Additionally, (10) suggests a linearization form of variance estimation of $\widehat{\theta}_I$. To estimate $\omega_i^\star = \pi(\boldsymbol{x}_i)^{-1}$, we can use employ the covariate balancing idea of Wong & Chan (2018) to obtain $\widehat{\omega}$ as provided in (7). The proof of Theorem 1 is presented in the Appendix.

## 4. Experiments

We conducted a simulation study to test our proposed method. Generally, our synthetic data consists of a super-population model and a propensity score model. We also

compare them in two scenarios with different superpopulation model: a linear case and a nonlinear case. In both cases, we keep the response rate around $70\%$ and $\mathrm{Var}(Y) \approx 10$. For the linear case, let $\boldsymbol{x}_i = (x_{i1}, x_{i2}, x_{i3}, x_{i4})^{\mathrm{T}}$ be generated IID element-wisely from uniform distribution on the support $(1, 3)$. The responses for linear case (Model A) are generated by

$$y_i = 3 + 2.5x_{i1} + 2.75x_{i2} + 2.5x_{i3} + 2.25x_{i4} + \sigma\epsilon_i,$$

where $\{\epsilon_i\}_{i=1}^n$ are generated from standard normal distribution and $\sigma = \sqrt{3}$. For the nonlinear case (Model B), we use

$$y_i = 3 + 0.2x_{i1}x_{i2}x_{i3}^2 + 0.3x_{i3}x_{i4} + \sigma\epsilon_i$$

to generate data with nonlinear structure.

As for the propensity score model, the response indicator variable $\delta$ are independently generated from Bernoulli distribution with probability $\mathrm{logit}(\mathbf{X}\boldsymbol{\beta} + 2.5)$, where $\mathbf{X} = [\boldsymbol{x}_1^{\mathrm{T}}; \ldots; \boldsymbol{x}_n^{\mathrm{T}}]$, $\boldsymbol{\beta} = (-1, 0.5, -0.25, -0.1)^{\mathrm{T}}$ and $\mathrm{logit}(p) = \log\{p/(1-p)\}$. We simulated data with sample size $n = 200$, $n = 500$ and $n = 1000$ with 1000 Monte Carlo replications. The RKHS we employed is the second-order Sobolev space.

We also compare different methods for imputation: KRR, B-spline, linear model (LM). The corresponding results are presented in the Table 1. In Table 1, the performance of the three imputation estimators are presented. In the linear case (Model A), the three methods show similar results. In the nonlinear case (Model B), KRR imputation shows the best performance in terms of MSE. Linear regression imputation still provides unbiased estimates, because the residual terms in the linear regression model are approximately unbiased to zero. However, use of linear regression model for imputation leads to efficiency loss because it is not the best model.

In addition, we have computed the proposed variance estimator under KRR imputation. The behavior of variance estimation for KRR is presented in Table 2. In Table 2, the RB $(\hat{V})$ denotes the relative bias of the proposed variance estimator. The relative bias of the variance estimator decreases as the sample size increases, which confirms the validity of the proposed variance estimator.

We also compute the confidence intervals using the asymptotic normality of the KRR imputed estimator. The proposed variance estimator is used in computing the confidence intervals. Table 3 shows the coverage rates of the confidence intervals. The realized coverage probabilities are close to the nominal coverage probabilities, confirming the validity of the proposed interval estimator.

*Table 1.* Biases, Variances and Mean Squared Errors (MSEs) of three imputation estimators

| Model | $n$ | | KRR | B-spline | LM |
|---|---|---|---|---|---|
| A | 200 | Bias | -0.0113 | 0.0027 | 0.0023 |
| | | Var | 0.0682 | 0.0679 | 0.0682 |
| | | MSE | 0.0684 | 0.0679 | 0.0682 |
| | 500 | Bias | -0.0032 | 0.0038 | 0.0038 |
| | | Var | 0.0263 | 0.0263 | 0.0263 |
| | | MSE | 0.0263 | 0.0263 | 0.0263 |
| | 1000 | Bias | -0.0037 | 0.0002 | 0.0002 |
| | | Var | 0.0128 | 0.0128 | 0.0129 |
| | | MSE | 0.0128 | 0.0128 | 0.0129 |
| B | 200 | Bias | -0.0011 | 0.0134 | 0.0117 |
| | | Var | 0.0674 | 0.0697 | 0.0697 |
| | | MSE | 0.0674 | 0.0699 | 0.0699 |
| | 500 | Bias | 0.0024 | 0.0114 | 0.0107 |
| | | Var | 0.0253 | 0.0262 | 0.0262 |
| | | MSE | 0.0253 | 0.0264 | 0.0263 |
| | 1000 | Bias | -0.0007 | 0.0071 | 0.0066 |
| | | Var | 0.0123 | 0.0127 | 0.0128 |
| | | MSE | 0.0123 | 0.0128 | 0.0128 |

*Table 2.* Relative biases of the proposed variance estimator under kernel ridge regression imputation

| Model | Sample Size | | |
|---|---|---|---|
| | 200 | 500 | 1000 |
| A | -0.0698 | -0.0376 | -0.0033 |
| B | 0.0593 | 0.0238 | 0.0120 |

*Table 3.* Coverage rates (%) of the proposed confidence intervals

| Model | Nominal Coverage | Sample Size | | |
|---|---|---|---|---|
| | | 200 | 500 | 1000 |
| A | 90 | 88.7 | 89.5 | 89.8 |
| | 95 | 94.3 | 94.6 | 95.1 |
| B | 90 | 88.1 | 89.5 | 89.3 |
| | 95 | 93.6 | 93.8 | 94.3 |

## 5. Conclusion

We consider Kernel ridge regression as a tool for nonparametric imputation and establish its asymptotic properties. In addition, we propose a linearized approach for variance estimation of the imputed estimator. Theoretical vailidity of our linearized variance estimator is established. Numerical studies confirm our theoretical results.

## Acknowledgements

The authors are grateful for very constructive comments of two anonymous reviewers. The research of the second author was partially supported by a grant from U.S. National Science Foundation (1931380).

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
