# OpenReview forum: "Variance estimation after Kernel Ridge Regression Imputation "
_ICML.cc/2020/Workshop/Artemiss — ICML Artemiss 2020_

### Official Review · AnonReviewer1 · 2020-06-18
**Good paper - exposition could be made clearer**

**Rating:** 7
**Confidence:** 4

**Review:**

This paper establishes $\sqrt{n}$-consistency and asymptotic normality of kernel ridge regression imputation under a MAR model and also shows how to estimate inverse propensity scores using covariate balancing (Wong & Chang 2018).

Overall, I found the results compelling and only have minor comments on exposition:
- From my understanding, this paper is specifically focusing on the goal of computing the average of the label (the $y_i$) values (averaged across feature vectors sampled the same way as training data), with some training label values missing. I'd suggest making this more obvious in Section 1 of the text. In particular, imputation is used in conjunction with many downstream tasks, and it would be helpful orienting the reader quickly as to what task your paper examines/what the target is that you're trying to estimate (and that KRR is only used as a subroutine).
- From a first read, it wasn't immediately obvious to me that the $\widehat{\omega}_i$ terms are the estimates of the inverse propensity scores. I would suggest making this more clear up front, possibly even before stating equation (6).
- I take it $\bar{\eta}$ is the mean of the $\hat{\eta}_i$ terms?
- While much of the focus is on KRR, if I understand things correctly, the covariate balancing approach can be used with other estimators as well that aren't KRR? It would be helpful making this clear earlier in the text, for example possibly even mentioning it in the abstract and/or Section 1. It comes off as a contribution that could be of independent interest.

---

### Official Review · AnonReviewer2 · 2020-06-24
**fractional imputation by kernel regularized regression**

**Confidence:** 4
**Rating:** 8

**Review:**

The paper is clear and present a novel fractionnal imputation method. However, some results of the simulation study should be more discussed :

- Authors write in the simulation study "the three methods show similar results" in configuration A,
but, KRR is 15 times more biased than B spline or linear model imputation (with 200 and 1000 individuals).
Furthermore, how can we explain constant bias between 500 and 1000 individuals for KRR, whereas this biais is highly smaller for others methods ? Relative bias  (as for variance estimation) would be helpful for interpretation of bias.

- In the introduction, authors position their work according to other fractionnal imputation methods.
They underline their approach alleviates the effects of parametric model assumption.
From this point of view, comparison with other kernel regression approaches would be expected.

- Authors highlight the coverage probabilities are close to the nominal coverage probabilities, that is right, but it is globally under the nominal level (which is significant with 1000 simulations). This undercoverage should be explained. Is it due to the uncertainty related to the estimate of lambda ?

---

### Decision · Program_Chairs · 2020-07-02

**Decision:**

Accept

**Comment:**

We are very happy to inform you that your paper has been accepted for the Artemiss workshop. We will contact you soon to inform you about the details concerning the format of your presentation at the workshop, and the camera-ready version deadline. Please take into account the referee's comments to write the camera-ready version.